# Cannabinoids’ Role in Modulating Central and Peripheral Immunity in Neurodegenerative Diseases

**DOI:** 10.3390/ijms25126402

**Published:** 2024-06-10

**Authors:** Nitzan Sharon, Ludmila Yarmolinsky, Boris Khalfin, Sigal Fleisher-Berkovich, Shimon Ben-Shabat

**Affiliations:** Department of Clinical Biochemistry and Pharmacology, Faculty of Health Sciences, Ben-Gurion University of the Negev, Beer-Sheva 8410501, Israel; nitzan.sharo@gmail.com (N.S.); yludmila@post.bgu.ac.il (L.Y.); boriskh83@gmail.com (B.K.); fleisher@bgu.ac.il (S.F.-B.)

**Keywords:** cannabinoids, neurodegenerative diseases, cannabinoid receptor, immunity

## Abstract

Cannabinoids (the endocannabinoids, the synthetic cannabinoids, and the phytocannabinoids) are well known for their various pharmacological properties, including neuroprotective and anti-inflammatory features, which are fundamentally important for the treatment of neurodegenerative diseases. The aging of the global population is causing an increase in these diseases that require the development of effective drugs to be even more urgent. Taking into account the unavailability of effective drugs for neurodegenerative diseases, it seems appropriate to consider the role of cannabinoids in the treatment of these diseases. To our knowledge, few reviews are devoted to cannabinoids’ impact on modulating central and peripheral immunity in neurodegenerative diseases. The objective of this review is to provide the best possible information about the cannabinoid receptors and immuno-modulation features, peripheral immune modulation by cannabinoids, cannabinoid-based therapies for the treatment of neurological disorders, and the future development prospects of making cannabinoids versatile tools in the pursuit of effective drugs.

## 1. Introduction

Cannabinoids constantly attract attention from clinicians, pharmacologists, and researchers because of their enormous therapeutic potential and ability to decrease inflammation and oxidative stress.

Diversity of origin allows separating cannabinoids into several groups: the endocannabinoids, the synthetic cannabinoids, and the phytocannabinoids [1]. The endocannabinoids are produced in the human body [2]. More than 450 synthetic cannabinoids are known [3]. Phytocannabinoids are compounds of plant origin; they were identified in the plant *Cannabis sativa*.

Various pharmacological properties of cannabinoids were reported [4]; among other things, they are effective anti-inflammatory or immune-suppressive agents, according to preclinical findings. Furthermore, cannabinoids are promising mediators for the treatment of neurodegenerative diseases, depression, cancer, and so on [5,6,7,8,9,10] according to clinical studies, but they have many undesirable side effects, for example, heart disorders, anxiety, possible loss of motor coordination, and impaired memory [4,11,12,13].

Knowledge and understanding of the expression, function, and regulation of cannabinoids and their receptors are important for developing novel approaches for controlling complex pathological processes induced by dysregulation of both the immune and the nervous systems. Although the role of cannabinoids in the treatment of neurodegenerative diseases was discussed in many reviews [6,7,8,9,10,14,15], their function in modulating central and peripheral immunity deserves particular attention. Several databases, namely PubMed, Google Scholar, Scopus, and Science Direct, were examined using key words; post-2001 publications were included which reflect the main facts, ideas, and concepts of the review. With this in mind, we aim to provide the best possible information about cannabinoids for the treatment of neurodegeneration and discuss cannabinoid receptors and immunomodulation features, peripheral immune modulation by cannabinoids, and future perspectives of cannabinoid-based therapies for the treatment of neurological disorders. Figure 1 demonstrates the crosstalk between the nervous and immune systems, as discussed in this review.

## 2. Cannabinoid Receptors and Immunomodulation

The endocannabinoid system (ECS) is an important multifunctional part of the nervous system and some peripheral tissues; its main components are cannabinoids, cannabinoid receptors (CBRs), and various proteins [2,4,6]. Its function is connected to numerous processes such as pain, food intake, energy metabolism, inflammation, immunity, and gastrointestinal, hormonal, and cardiovascular activities, and so on [2,3,4,5,6].

The two main cannabinoid receptors are CB1Rs and CB2Rs; they are G-protein-coupled receptors that participate in various physiologic processes. The cannabinoids’ psychoactive effects are mediated by CB1R, which was discovered in 1988 by Howlett’s group [16]. In 1993, CB2R was identified in the rat spleen, characterized as a peripheral cannabinoid receptor [17] and found mainly in immune cells and tissues [17]. Figure 2 demonstrates the expression of CB1R and CB2R in the nervous and immune systems. G-protein-coupled receptor 55 (GPR55) is the endocannabinoid system’s third main receptor. It was identified in 1999 and found to be extensively expressed in the human brain [18] but also in peripheral tissues [19,20]. Its role is still unclear, but accumulative evidence indicates that GPR55 has physiologic roles [19,20]. Cannabinoids act via other targets, e.g., transient receptor potential (TRP) channels [21] and nuclear receptors such as peroxisome proliferator-activated receptors (PPARs) [22]. Here, we describe the immunomodulatory roles of cannabinoid receptors CB1R, CB2R, GPR55, and PPARγ in the central nervous system (CNS) and periphery.

### 2.1. Cannabinoid 1 Receptors in the CNS

CB1 receptors (CB1Rs) are abundantly expressed G-protein-coupled receptors in the CNS. CB1Rs are particularly abundant in the frontal cortex, hippocampus, basal ganglia, hypothalamus, cerebellum, and spinal cord. CB1Rs are expressed prominently by most neurons and at a lower level by immune cells [23]. Their expression is variable in neurodegenerative diseases. CB1Rs are upregulated in the early stages of Alzheimer’s disease (AD) and found to be downregulated during the progression of the disease throughout different brain regions, including the cortex, the amygdala, the hippocampus, and the cerebellum [24]. In mice, deletion of CB1Rs has been associated with cognitive and memory decline, loss of neurons in the hippocampus, enhanced glial cell activity, and upregulated levels of tumor necrosis factor-alpha (TNFα). It was suggested that the upregulation of CB1s in the brain of AD patients can be used as a therapeutic target [25]. In Huntington’s disease (HD), CB1Rs are downregulated in the striatum; this is one of the earliest changes in HD pathogenesis. This reduction occurred in all models of animal HD in humans and correlated with the progression of HD [26]. In postmortem brain tissues from multiple sclerosis (MS) patients, CB1Rs were abundantly expressed in cortical neurons and neurons in white matter areas, especially demyelinated areas. Also, in active plaques, adult oligodendrocytes, oligodendrocyte precursor cells (OPCs), and microglial cells express CB1Rs [27,28].

It seems that CB1Rs have an important role in MS progression. Genetic deletion of CB1Rs in mice led to more severe disease than in WT mice in experimental autoimmune encephalomyelitis (EAE), an in vivo disease model of MS [29]. Over-expression of CB1Rs delayed the onset and improved the severity of the disease in EAE. This was accompanied by the production of neurotrophic factors in the spinal cord, which are important to the development and function of neurons [30]. In Parkinson’s disease (PD) patients, CB1 receptors were downregulated in the substantia nigra and striatum in the presymptomatic stages and upregulated as the disease progressed. Moreover, PD medication such as levodopa have been shown to increase the expression of CB1Rs [31,32].

CB1 modulates and implicates wide molecular mechanisms that relate to neurodegenerative processes. These mechanisms include decreasing excitotoxicity, shifting polarization of microglial cells, promoting neuronal survival, and suppressing pro-inflammatory factors [23].

CB1Rs’ activation promotes the inhibition of adenylate cyclase, inducing a decrease in the level of second messengers such as cyclic adenosine monophosphate (cAMP), an event related to a decrease in glutamate release. CB1Rs are present in glutamatergic synapses and, while activated, regulate excessive glutamate release, leading to excessive Ca^2+^ influx, thus reducing excitotoxicity and oxidative stress and promoting neuronal survival. These neuroprotective effects lead CB1Rs to be considered therapeutic targets of neurodegenerative diseases in which glutamatergic disruption is related to disease pathology [33,34]. CB1R deletion on glutamatergic neurons exacerbates quinolinic acid-induced excitotoxic damage and striatal neurodegeneration in HD mice [35,36].

Also, GABAergic signaling alterations contribute to neurodegenerative and neuroinflammation [37]. Neurons expressing CB1Rs in the cortex and the hippocampus are vastly GABAergic neurons. Lipopolysaccharide (LPS)-treated GABAergic neuron-specific CB1R knockout mice (GABA/CB1−/−) express pro-inflammatory microglial phenotype (M1). GABAergic neurons have a role in the regulation of microglial activity. Altered expression of neuron–glia interaction was reported in hippocampal GABAergic neurons in GABA/CB1−/− mice. These indicate that CB1R agonists can modulate microglial activity indirectly through CB1Rs on GABAergic neurons [38].

CB1Rs can regulate microglial activity directly since these receptors are expressed in microglia and were shown to be upregulated in activated microglia [39,40]. It is important to mention that microglia express CB1Rs at a low level but still influence microglial activity [40]. For example, deletion of CB1Rs in mouse microglial cells led to a reduced reactivity to LPS in a sex-specific manner [41]. Inhibition of CB1Rs in BV2 microglial cells induced M1 phenotype and promoted the secretion of pro-inflammatory mediators such as TNF-α, interleukin (IL)-1β, and IL-6 while inhibiting the production of anti-inflammatory cytokine IL-10 and chemokines [30]. In mice, in CX3CR1-microglia, in which CB1 was abolished, exposure to immune challenge reduced pro-inflammatory cytokines in the CNS [41].

On the other hand, some studies reported that upregulation and activation of CB1Rs lead to neuroprotection and anti-inflammatory responses. Acute activation of CB1Rs in the hippocampus of rats prevents the reactive gliosis induced by amyloid beta peptide (Aβ), neuronal degeneration, and the production of nitric oxide (NO) [42]. Activation of CB1Rs attenuates rotenone and alpha-synuclein-dependent inflammation in neurons and astrocytes in SH-SY5Y and C8-D1A cell culture models [43]. In astrocytes, activation of mitochondrial CB1Rs reduces the generation of reactive oxygen species (ROS) in mouse models [44].

CB1Rs promote extracellular signal-regulated kinase 1/2 (ERK1/2) activation [45,46,47]. This pathway regulates microglia’s inflammatory gene expression and immune responses and has been linked to neuroinflammation and neurodegenerative diseases [48,49]. For example, in the EAE model, inhibition of ERK activation reduces the severity of the disease. This effect was associated with suppression of autoantigen-specific Th17 and Th1 responses [50]. Also, ERK activation via the protein kinase B pathway promotes neuronal differentiation triggered by CB1R signaling [47]. This contributes to its neuroprotective activity. Indeed, in an in vitro model of HD, activation of CB1Rs protects cells from death, mediated by phosphorylation of ERK [46].

#### Cannabinoid 1 Receptors in the Periphery

Although CB1Rs were found mainly in the CNS, CB1Rs also have functional roles in immunomodulation in the periphery. In the periphery, CB1Rs are expressed in tissues like the GI tract, lungs, and skin [24,51] and in cells relating to metabolism and the reproductive and immune systems [23,52].

CB1Rs are immunomodulators in macrophages [51,53], T cells [52,54], neutrophils [55], mast cells [56], and dendritic cells [57].

Activation of CB1Rs suppressed M2 macrophage differentiation indicated by downregulation of the markers IL-10, CCL22, Arg-1, and CD206 [51] and promotes M1 differentiation [58]. Also, it has been reported that CB1 promotes pro-inflammatory responses of macrophages through ROS production [53]. Furthermore, blocking CB1Rs has anti-inflammatory effects on macrophages as downregulation of pro-inflammatory cytokine expression and phagocytosis [59]. Interestingly, splenocytes from mice deficient in CB1Rs respond strongly to LPS stimuli and secrete pro-inflammatory cytokines at higher levels than WT mice [60]. In splenic T cells and splenocytes from mice that were immune-activated by *Legionella pneumophila*, activation of CB1Rs suppressed IL-12Rβ2 (neuroprotective receptor) expression [61]. Also, CB1Rs are involved in neutrophil activation, and CB1R blockage suppresses their entry to injured areas and attenuates peripheral inflammation [55,62,63]. In mast cells, CB1R activation leads to sustained elevation in cAMP levels and reduction in mast cell activation [56]. In LPS-activated dendritic cells, CB1R activation reduced CD83 expression and TNFα and IL-6 production [57].

### 2.2. Cannabinoid 2 Receptors

CB2Rs are found mainly in peripheral immune cells and immune organs like the spleen, tonsils, and thymus and in relatively low levels in non-immune tissues such as reproductive tissues and the lungs [17].

In the CNS, CB2Rs are expressed in glial cells, primarily in microglial cells, and the expression is upregulated under inflammatory conditions [64]. CB2Rs are also expressed in neurons in different brain areas and appear to have regulatory functions for neuronal activity. Increased expression of CB2Rs was reported in pathological conditions, where neuroinflammation is enhanced, mostly in microglial cells. Upregulation of receptors was reported in AD [65], PD [27,66], MS [27], HT [67], and amyotrophic lateral sclerosis (ALS) [68]. Lopez et al. (2018) reported that in a mouse model of AD, 5xFamilial AD, mouse CB2Rs were upregulated in areas of neuroinflammation where microglial cells are activated and in areas of plaque formation where amyloid is present. The increase in CB2R expression was detected when amyloid plaques appeared. This suggests that CB2R levels increased after prolonged neuroinflammation and that CB2Rs can be a marker of neuroinflammation and early AD pathology. Genetic deletion of CB2Rs in this model causes a decrease in amyloid plaques [65]. In another study, genetic deletion of CB2Rs decreased microglial activation and harmed their ability to phagocytose Aβ. Aβ accumulation can trigger the activation of CB2Rs in microglial cells, as shown by p38 activation [69]. CB2Rs induce Toll-like receptor (TLR)-mediated microglial activation, and it is mainly p38 dependent. Deletion of CB2Rs leads to the loss of TLR functions and changes in microglial gene expression and morphology [70]. TLR promotes a signaling cascade that eventually elevates the production of cytokines, chemokines, and ROS [71]. In mice lacking CB2Rs, macrophages stimulated with LPS/interferon-gamma (IFN-γ) activate the release of cytokines like TNF-α, chemokine (C-C motif) ligand 2, interleukin 6, and other inflammatory markers. Also, gene signatures for cytokine secretion were downregulated in primary microglial cells from mice deficient CB2Rs [70].

CB2R deletion improved cognitive and learning deficits and decreased microglial activation and amyloid levels in amyloid precursor protein transgenic mice [72]. Also, deleting CB2Rs reduced neuroinflammation and migration of macrophages to the CNS [73]. Conversely, pharmacological CB2R antagonists do not affect microglial responses to LPS/IFN-γ as observed in CB2 genetic deletion mice. It is likely to assume that CB2Rs have a major role in microglial development and functions, and deletion of CB2Rs leads microglia to be immune to inflammatory stimuli [74].

By contrast, several studies showed that a lack of CB2Rs causes inflammatory responses [17]. For example, microglia from CB2R-deficient mice fail to polarize toward the anti-inflammatory M2 phenotype [75]. In EAE studies, CB2R deficiency aggravated EAE symptoms, CD4+ T-cell infiltration, and microglial activation [76]. Lack of CB2Rs in mice that express human mutant huntingtin exon 1 increased microglial activation, exacerbated disease, and reduced mouse lifespan [67]. In astrocytes, the knockdown of CB2Rs aggravated motor function and led to the death of neurons [77].

In microglia, activation of CB2Rs shifted the M1 phenotype to the M2 phenotype [78] and reduced secretion of pro-inflammatory cytokines and mediators like inducible nitric oxide synthase (iNOS) while increasing production of anti-inflammatory mediators and decreasing immune cell infiltration into the CNS [64,76,77,79,80,81,82]. Moreover, activation of CB2Rs showed neuroprotective effects in models of PD [77,82,83], HD [67], ALS [68], MS [76,84,85,86], and AD [64,87]. In EAE mice, CB2R activation ameliorated clinical signs of disease severity, reduced neuroinflammation and demyelination in the spinal cord, and elevated anti-inflammatory cytokines [76,84,85,86]. In a mouse model of spinal cord injury (SCI), activation of CB2Rs prevented loss of myelin, neuron death, and gliosis [78]. In a PD animal model, CB2R activation protected dopamine neurons from degeneration, reduced blood–brain barrier (BBB) damage, infiltration of peripheral immune cells, and production of inflammatory mediators by activated microglia [77]. Indeed, CB2Rs seem to play important roles in recruiting peripheral immune cells into the CNS [88]. CB2Rs limited the release of chemokines, adhesion molecules, and chemokine receptors in T cells, decreased neutrophil, T-cell, and macrophage infiltration, and reduced permeability of the BBB [76,77,82,84,85,89]. Besides the effect on trafficking immune cells, CB2R activation suppressed the cytotoxic activity of CD8+ T cells and balanced differences in CD4+ T cells, T helper (Th) 17 and Th1 to Th2, and regulatory T cells (Treg) [17,84,90,91,92].

CB2Rs regulate cytokines and ROS production, cell proliferation, apoptosis, and phagocytosis [17,53,54,90,92,93,94]. These effects occur via signal transduction pathways such as adenylyl cyclase inhibition, p38 and ERK1/2 activation, and nuclear factor kappa B (NF-κB) [17,70,94]. In the periphery, these effects make CB2 an immunoregulator of pathologic conditions that involve inflammation such as psoriasis [95], gastrointestinal (GI) inflammatory diseases [96], viral infections [97,98], arthritis [99], and asthma [100].

### 2.3. G-Protein-Coupled Receptor 55 (GPR55)

GPR55 is a G-protein-coupled receptor considered as a third cannabinoid receptor. It is highly expressed in the CNS, in the frontal cortex, striatum, hippocampus, and cerebellum in neurons and microglial cells. It is also expressed in the periphery, in tissues like GI, liver, spleen, and immune cells [19,101]. GPR55 leads to the activation of transcription factors, such as the nuclear factor of activated T cells (NFAT) and NF-κB [19] and the activation of Mitogen-activated protein kinases (MAPK), such as p38 and ERK1/2 [102]. Also, an increase in intercellular Ca^2+^ was reported following GPR55 activation [103].

GPR55 has been shown to be involved in inflammation and neuroprotection [101,102,104]. Upregulation of GPR55 was reported in AD mice upon the accumulation of Aβ plaques, as shown in the APP knock-in AD mouse model. GPR55 was detected, especially in microglia surrounding the Aβ plaques [105]. In other AD in vivo studies, activation of GPR55 attenuated cognitive impairment [106], prevented microglial activation, and reduced the hippocampus’s TNF-α, IL-1β, and IL-6 expression while increasing IL-10 expression. Reduced oxidative stress and neuronal apoptosis by inhibiting NF-κB were shown in LPS-treated mice [107]. GPR55 in the brain is expressed highly in areas related to motor functions, so it is considered a target for diseases like PD. Indeed, in the chronic MPTP mouse model of PD, GPR55 was downregulated in the striatum, and activation of GPR55 improved motor impairments and protected dopaminergic neurons [101]. Pharmacological activation of GPR55 led to neuroprotection effects in other mouse models of PD and to a reduction in microglial activation and expression of pro-inflammatory cytokines and enzymes [108]. Neural stem cells treated with GPR55 agonists showed increased mRNA levels for anti-inflammatory or neuroprotective cytokine receptors (IL-10Rα, TNFR2), while inflammatory cytokine receptor mRNA expression was downregulated. Moreover, mice deficient in GPR55 showed more prolonged and robust inflammatory responses to chronic systemic exposure to LPS. Interestingly, microglial activation did not change after agonist intrahippocampal administration [109]. Other studies indicate microglia mediate these neuroprotective effects of GPR55. In vitro, activation of GPR55 exerted microglia-dependent neuroprotection [104] and suppression of NO, ROS, IL-6, and phagocytic activity by LPS-stimulated BV-2 microglial cells [110]. In contrast, antagonism of GPR55 in primary microglial cells attenuated microglial activation and secretion of Prostaglandin E2 and Cyclooxygenase-2 [102].

In the periphery, the GPR55 antagonist limited leukocyte migration and activation in a model of intestinal inflammation [111] and suppressed the inflammatory response in a sepsis model [93] and endothelial cell model [112]. In stimulated monocytes and NK cells, activation of GPR55 increased pro-inflammatory mediator release and impaired the phagocytotic ability of monocytes [113]. In peripheral blood mononuclear cells, GPR55 levels were elevated during activation in various models [114], as well as in patients with Crohn’s disease and ulcerative colitis (UC) [115].

### 2.4. Peroxisome Proliferator-Activated Receptor Gamma (PPARγ)

PPARγ is a nuclear receptor that regulates cellular functions, energy homeostasis, glucose, and lipid metabolism. The isoform PPARγ1 is expressed widely in human tissues like the spleen, heart, colon, and immune cells, and PPARγ2 is expressed mainly in adipose tissue [116,117].

Accumulating evidence indicates that cannabinoids (endocannabinoids, phytocannabinoids, and synthetic cannabinoid ligands) activate PPARγ receptors and can modulate processes like immune response, analgesia, apoptosis, vasocontraction, and metabolism [22]. Cannabinoids, including CBD (cannabidiol), THC (Δ-9-tetrahydrocannabinol), and THCA (tetrahydrocannabinolic acid), have been reported to exert neuroprotective and anti-neuroinflammatory effects in a PPARγ-dependent manner in glial cells and neurons [118,119] in in vitro and in vivo models of HD [120], PD [108,121], and ALS [122]. Non-cannabinoid ligands of PPARγ showed anti-inflammatory effects in microglia and astrocytes and in in vivo models of MS [123], AD [124], PD [125], ALS [126], and HD [127].

The expression of PPARγ was found to be altered in pathologic conditions [116,117,123,128]. For example, during active EAE, PPARγ is upregulated in microglia and astrocytes [123]. Downregulation of PPARγ was reported in the monocytes of relapsing/remitting MS patients. The inflammatory environment in MS includes high levels of IFNγ, IL1β, and myelin protein. This decreased the expression of PPARγ in macrophages [129].

PPARγ is an important modulator in inflammatory responses as it inhibits the expression of pro-inflammatory mediators by inhibiting transcription factors such as NF-κB [116]. Deletion of PPARγ in colonic epithelial cells and immune cells led to an increase in pro-inflammatory mediators and adhesion molecules and exacerbated colitis and inflammatory bowel disease (IBD) in vivo [116]. In colonic biopsies from UC patients and mice, PPARγ activation decreased inflammatory cytokine release [130].

Interestingly, it was discovered by bioinformatics analysis that the gene that encodes for PPARγ (PPARG) was significantly decreased in both AD and UC [131]. PPARγ promotes the M2 phenotype in macrophages and microglia [117,132]. It promoted phagocytosis of amyloid beta [124] and myelin [133]. Thus, PPARγ may be a potential target for neural pathology induced by peripheral inflammation and vice versa.

Figure 3 illustrates that physical activity, stresses, food consumption, sexual behavior, orgasm, obesity, inflammation, tissue damage, and other stimuli may trigger the release of endocannabinoids, and their high levels cause various consequences.

## 3. Cannabinoids in Central Immunity and Neurodegenerative Diseases

The interaction between the brain and the immune system is complex and bidirectional. This interaction occurs through various neural, hormonal, and molecular communication pathways. The brain and the immune system communicate extensively, influencing each other’s function and response to various internal and external stimuli. This intricate interaction is crucial for maintaining homeostasis and responding effectively to challenges such as infection, injury, and stress.

The immune cells are in the borders of the brain (the choroid plexus, meninges, and perivascular spaces); therefore, they interact with the brain distantly with the involvement of the blood vasculature, lymphatic system, and skull microchannels [134].

Microglia participate in various processes, including free radical reduction, cytokine and chemokine secretion, phagocytosis of debris, steroid release, repair of cells, and so on [135]. The microglia may be pro- and anti-inflammatory [136].

M1 microglia may produce pro-inflammatory cytokines (interleukins (IL-1 and IL-6), TNF-α, and IFN-γ) and reactive oxygen species and may damage healthy neurons, causing synaptic dysfunction, loss of synapses, and neuronal death. Neuroinflammation, which involves the activation of microglia and astrocytes, produces pro-inflammatory cytokines, which can activate a series of pathways [137]. Figure 4 briefly depicts the pathways of cannabinoid effects.

In addition, neuroinflammation contributes to the onset and progression of neurodegenerative diseases [138]. A common feature of these diseases is reduced hippocampal neurogenesis [139], which is poorly understood.

Unfortunately, neurodegenerative diseases currently cannot be cured.

Their mechanisms of protection are complex and involve receptors in the brain but also in immune cells; many preclinical studies demonstrated the excellent potential of various cannabinoids in the case of neurodegenerative diseases [4]. They are effective neuroprotective compounds because they engage both receptors in the brain and immune cells [140]. However, there are many concerns about developing safe, effective drugs, including insufficient knowledge of the modes of action of these compounds, their optimal dosage, short- and long-term impacts, and side effects. For example, CBD demonstrated significant anti-amyloidogenic, antioxidative, anti-apoptotic, anti-inflammatory, and neuroprotective properties [141], but its numerous toxic properties were reported [142].

Cannabinoids are administrated by different routes in neurodegenerative diseases, including transdermal administration, oral administration, mucosal administration, and subcutaneous administration [143]. Modern nanotechnologies may enhance the bioavailability of these compounds [144].

### 3.1. Parkinson’s Disease

PD is a chronic neurodegenerative age-related condition of the central nervous system with motor and nonmotor symptoms. Morphological basis is characterized by misfolded α-synuclein seeds, which are collected in fibrillary inclusions (Lewy bodies and Lewy neuritis) and by shedding of dopamine neurons in the substantia nigra [145]. Although the underlying molecular mechanisms of PD are not entirely clear, mitochondrial DNA defects, bioenergetic disorders, ROS generation, and dysfunctional calcium homeostasis increase neuronal death [146].

In PD, pain is a frequent, debilitating, and often neglected non-motor symptom for which there is no truly effective treatment [147].

This disease is estimated to affect at least 4.1 million people globally, and the numbers of patients with PD is predicted to double by 2030 [148].

The emphasis shifts from symptomatic treatment to reducing the progression and/or curing PD. Patients with PD mainly obtain only symptomatic treatments that are aimed at correcting motor disorders without eliminating the cause of the disease. Levodopa is the most widespread standard drug for treating this disease [149]. Although this drug is considered the most effective [150], long-term consumption causes various complications in the long term and is not effective [151].

Some cannabinoids are effective for treating PD on the one hand; on the other hand, their harmful and adverse effects are widespread. The cannabinoid system featured in the basal ganglia greatly influences the progression of PD [152].

Table 1 demonstrates the potential efficacy of cannabinoids in the treatment of PD; preclinical findings mark improving motor functions and neuroprotective effects; unfortunately, clinical studies of cannabinoids remain limited. For example, seven patients with PD had significant improvements in measures of functioning and well-being after treatment with CBD 300 mg/day compared to a placebo group [153]; it was reported that CBD decreased psychotic symptoms [154].

### 3.2. Alzheimer’s Disease

More than 5% of the population aged 60 and older suffers from AD [169]. The reduced levels of choline acetyltransferase and an extracellular deposit of β-amyloid plaques and neurofibrillary tangles composed of hyperphosphorylated tau protein are pathogenetic features of this disease [170].

Many cannabinoids may interact specifically with the Aβ, but CBD and THC are much better studied in this relation [171].

THC demonstrated anti-amyloid aggregation activity in an in vitro study [172]. In addition, many THCs influenced Aβ fibril formation and aggregation [173]. This cannabinoid promoted the destruction of intracellular Aβ and inhibited the inflammatory reaction [174]. THC blocked acetylcholinesterase activity [175]. Cognitive functions in old mice were improved following the treatment with a low dose of THC [176].

The neuroprotective, anti-inflammatory, and antioxidant properties of CBD, its ability to prevent hippocampal and cortical neurodegeneration, and its regulation of migration of microglial cells were reported [177].

Unfortunately, all clinical studies were short (maximum 14 weeks). For example, one such study demonstrated that 1.5 mg of THC per day enhanced the mobility of the patients without any adverse effects during two 1-week periods [178].

As a result, it is not clear whether cannabinoids are beneficial or harmful for patients with AD [15,179]. Further preclinical and clinical studies should be performed to fully explore the therapeutic potential of each cannabinoid.

### 3.3. Multiple Sclerosis

MS is a non-traumatic disease of the central nervous system that is widespread among young adults; although the cause of MS is unknown, a combination of genetic and environmental factors (nutrition, Epstein–Barr virus infection, and smoking) is believed to be the reason why MS affects some people [180]. Significant demyelinating plaques are observed in grey and white matter in MS [181].

Taking into account that neurodegeneration is the main point in the pathogenesis of MS [182], many cannabinoids have a neuroprotective potential for MS treatment [183].

The preclinical studies showed the effectiveness of CBD in the inhibition of the onset and progression of MS [184]. This cannabinoid can also modulate Th17 reactions by decreasing IL-17 and IL-6 secretion and promoting IL-10, according to in vitro research [185]. In addition, it improved the clinical picture and decreased cellular infiltration and tissue damage in the central nervous system when the EAE model was applied [186].

Cannabigerol (CBG) was tested as a neuroprotective and anti-inflammatory agent both in cell cultures and in a mouse model of MS and EAE; TNF-α was significantly reduced, and an improvement in EAE symptoms was observed [187].

It was reported that two novel synthetic cannabinoids (based on the carbon–silicon switch strategy) were tested in *an* in vivo *model of MS; they significantly decreased the* infiltration of inflammatory cells into the brain and inhibited demyelination [188].

Clinical studies showed that nabiximols, *Cannabis* extract, and synthetic THC provide significant pain relief in patients with MS [189]. Δ9-THC reduced MS pain.

An administration of an oromucosal spray that contains THC and CBD in a 1:1 ratio led to a decrease in spasticity and pain relief in patients with MS, according to the results of clinical research [190].

In addition, several clinical studies demonstrated that nabiximols, in combination with other anti-spasticity medications, are effective for patients with MS when treatment continues from 6 weeks to 14 weeks [189].

## 4. Peripheral Immune Modulation by Cannabinoids

Cannabinoids have a broad range of activities involving the cells of the adaptive and innate parts of the immune system.

Cannabinoids regulate many functions of immune cells, for example, T helper cell development [191]. This modulation takes place through cannabinoid receptors and influences the secretion of cytokines and other products of the genes. Of note, THC reduces Th1 biasing activity, including IL-12Rbeta2, through a CB1-mediated mechanism and increases Th2 biasing activity, including GATA3, via the CB2 mechanism [192]. The conjunction of immune suppression and neuroprotection is one of the possible mechanisms by which cannabinoids can decrease inflammatory processes via CB1 [193].

Interestingly, the impact of THC on pregnant mice triggered epigenetic changes and long-term effects on the immune system in their offspring [194]. CBD caused a reduction in IL-17 and IFN-γ when a murine model of MS was used. As a result, T-cell infiltration into the central nervous system was decreased, but the production of myeloid-derived suppressor cells (MDSCs) was significantly higher than in the control [186].

It was shown that cannabinoids affect microglial microRNAs and their biochemical and epigenetic properties [195].

It was confirmed that the immune system plays an essential role in pain processing via releasing mediators that sensitize some specialized sensory neurons [196].

### Pain Relief

Many cannabinoids were reported as beneficial analgesic medications in treating the following pain conditions: chronic pain, fibromyalgia pain, and geriatric pain [197].

Chronic pain is characterized by long-lasting pain for many months and possibly years, even after the healing of an injury [198]; about 30% of the whole population worldwide suffers from this type of pain [199]. CB1Rs are associated with pain’s sensory and affective components because they are situated at key nodes on the path of pain [200]. Regarding the role of microglia in chronic pain, they produce endocannabinoids, express CB2Rs, and respond to cannabinoid agonists. Moreover, several complicated interactions exist between neurons, microglia, and astrocytes [201]. It is not clear how cannabinoids may modulate microglial function. Understanding these mechanisms may be important for using the full potential of cannabinoids for chronic pain treatment.

In studies of cannabinoid ligands in animal models of acute and chronic pain, many cannabinoids were tested only in animal models of acute pain; in this case, Δ9-THC, CBD, CP55,940, and WIN 55,212-2 were effective analgesics [202].

According to the clinical trials, cannabis with a THC concentration of at least 10% or lower had beneficial results in the treatment of chronic pain [203]. Nabilone was also tested clinically on patients with chronic pain; whether or not this synthetic cannabinoid is effective remains to be determined, but euphoria, drowsiness, and dizziness were the most frequent side effects of the drug [204]. An interesting fact is that nabilone relieved fibromyalgia pain [205].

## 5. Cannabinoid Impact on Peripheral Immunity in Neurodegenerative Diseases

There is much evidence to connect peripheral inflammatory diseases to the development of neurodegenerative diseases [206,207,208,209,210].

The gut microbiome plays a significant role in the progression and severity of degeneration and neuroinflammation. The microbiome influences BBB development and the production of inflammatory mediators [211,212]. Changes in the composition of the microbiome are reported when neurodegeneration is present. *Akkermansia muciniphila*, a bacterium, was reported to be increased in the microbiome of MS patients and animal models, with a correlation between exacerbations and the severity of the disease [213]. In contrast, the protective effects of *A. muciniphila* were reported on cognitive deficits and amyloid pathology in a mouse model of AD [214]. In the ALS model, the administration of *A. muciniphila* ameliorated ALS symptoms and improved mouse survival [215]. In addition, alteration in oral microbiota and periodontal disease can be a factor in neurodegeneration. In particular, *Porphytomonas gingivalis* is an important bacterial pathogen associated with sporadic AD [216]. *P. gingivalis* can cross from the periphery into the brain through the BBB and accelerate neuroinflammation, plaque formation, and neuronal death [217,218]. CBD, THC, and cannabinol (CBN) suppressed *P. gingivalis*-induced IL-12 p40, IL-6, IL-8, and TNFα release while enhancing the anti-inflammatory cytokine IL-10 from human innate cells [219].

Short-chain fatty acids (SCFAs) are common metabolites from the breakdown of dietary fiber by intestinal bacteria. SCFAs have immunomodulatory functions related to gut/immune homeostasis [220]. Recent studies suggest that there is a link between SCFAs and diseases such as obesity, diabetes, and neurodegenerative diseases [220]. Cannabinoids can affect the microbiome, SCFA composition, and immune cell population and thus may contribute to immunomodulation in the CNS. In the EAE model, CBD + THC treatment reduced disease severity and altered the gut microbiome. CBD + THC treatment led to reduced levels of *A. muciniphila* and increased the level of anti-inflammatory short-chain fatty acids such as butyrate [221]. In addition, endocannabinoid AEA treatment enhanced the abundance of beneficial bacteria-producing butyrate [222]. Butyrate is an important SCFA that participates in brain development and the progress of several diseases, including AD [223]. Also, butyrate modulates immune cells, leading to anti-inflammatory effects in the periphery and CNS [224].

It is reported that THC promotes the growth of bacterial species that produce neuroprotective metabolites in the microbiota, such as Indole Propionic Acid (IPA), which are significantly decreased in diseases like HD and AD [225,226,227]. IPA is considered to play a role in mediating between the gut and brain [228]. In other studies, THC reduces levels of *A. muciniphila* and increases beneficial bacteria populations and SCFAs such as propionic acid (PPA) [227]. PPA can cross the BBB into the brain, thus contributing to gut–brain communication. Studies showed that PPA promotes gliosis and cytokine release [229,230], but others implicated its anti-inflammatory effects [224,227,231,232]. For example, in human stem cells, PPA upregulates TNFα and anti-inflammatory cytokine IL-10 [229]. Interestingly, serum and feces samples from MS patients presented significantly reduced amounts of PPA compared with controls, particularly after the first relapse. Moreover, PPA supplementation intake increases regulatory T (Treg) cells, while Th1 and Th17 cells decrease in the gut and lead to fewer relapse events per year, improvement in disability, and reduced brain degeneration [232]. In the PD model, CBD treatment improves motor deficits and prevents αSyn accumulation in the brain, while in the feces, CBD reverses pathological metabolites and enzymatic products compared to untreated mice. CBD treatment downregulates metabolites in arginine biosynthesis and histidine metabolism, such as N-Acetyl-L-glutamate, capric acid, and 3-methylhistidine (3-MH). An increased level of 3-MH is a marker of AD in the CSF database and is also linked to metabolism dysfunction related to PD [233]. Capric acid is a medium fatty acid that showed neuroprotective effects in the PD model [234].

Chronic intestinal inflammation contributes to neurodegeneration. For example, IBD is a chronic disease characterized mainly by intestinal inflammation, and patients with IBD have a higher risk of PD and AD [208,235,236]. PPARG and NOS2 (encodes for iNOS) are shared genes in AD and UC, impacting macrophages and microglial polarization states [131]. PD patients suffer a high prevalence of GI symptoms, and gut inflammation has been described in PD [237]. Moreover, some studies propose that the onset of α-syn misfolding in the intestine is accompanied by inflammation and oxidative stress and later by α-syn spreading into the CNS [206]. Cannabinoids such as CBD and CBG suppress inflammation in the GI system [238]. In the EAE model, mice treated with CBD showed significant suppression of inflammation in the GI and intestinal epithelial cells, along with attenuation of EAE and regulation of inflammatory mediators in CNS [239].

Enteric glial cells (EGCs) have a key role in GI inflammation. EGCs are peripheral glial types in the enteric nervous system (ENS) and are often called “the second brain”. EGCs function in intestinal epithelium homeostasis, motility, and the immune and neuroimmune response. Over-activated EGCs are associated with degeneration, inflammation, and diseases such as PD and AD [206]. Phytocannabinoids can regulate EGC activity and have anti-inflammatory effects on EGCs. G. Cohen et al. 2023 reported that THC 0.1 μg/mL exerted an anti-inflammatory effect on EGCs, evaluated by reducing TNFα levels and diminishing glial fibrillary acidic protein (GFAP). Cannabichromene (CBC), CBG, THCA, and THC-COOH also showed anti-inflammatory effects in EGCs [240]. In addition, a positive correlation was found between S100B protein levels and the severity of motor symptoms in PD patients [241]. S100B protein expression in EGCs has been decreased by phytocannabinoids [240]. In the AD model, endocannabinoid palmitoylethanolamide (PEA) treatment regulated EGC pro-inflammatory responses, reduced Aβ and α-synuclein protein accumulation, and decreased TLR-4, IL-1β, and S100B protein expression levels in the colonic tissue of mice [242].

Immune cells in the periphery are affected by cannabinoids, and these effects can influence the CNS. Early oral administration of CBD, before the clinical signs or neuroinflammation, protected EAE mice from severe disease, both peripherally and in the CNS. Attenuation of neuroinflammation correlated with significant suppression of MOG35–55-specific IFN-γ-producing CD8+ T cells in the spleen [29]. CD8+ T cells mediate CNS autoimmune disease and can initiate severe CNS autoimmunity [243]. Thus, CD8+ T-cell suppression may explain these outcomes in the CNS. In another EAE study, CBD treatment at clinical sign onset significantly diminishes the severity of EAE. Less CD4+ and CD8+ T cell infiltrate is seen in CBD-treated mice’s spinal cords and brains. In the spleen, CD4+ T cells from CBD-treated mice secreted less inflammatory cytokines (IFNγ and IL-17) and increased anti-inflammatory cytokine IL-10 secretion [186]. Lenabasum, a synthetic CB2 agonist, significantly reduces the production of TNFα, IL-2, IFN-γ, and IL17 by T cells in relapsing/remitting MS patients [85]. Also, selective CB2 agonists such as O-1966, Gp1a [84], and JWH-133 [79,84] inhibited the infiltration of peripheral immune cells. HU30, a potent CB2 agonist, reduces the expression of chemokine ligands and receptors in the spinal cord and bone marrow of EAE mice [76]. CBD treatment promotes MDSCs in the CNS and periphery in the EAE model [239]. These cells can suppress T-cell activity, thus limiting inflammation. Indeed, in the EAE model, limiting MDSCs in the spleen are correlated to the severity of the clinical course, demyelination, and axonal damage [244]. Combined treatment with THC+CBD (1:1 ratio) also reduced disease severity in the EAE model while downregulating IL-17A and IFN-γ secretion in splenocytes and upregulating MDSCs compared to disease controls [221]. Table 2 summarizes the experimental data for neurodegenerative diseases in animal models.

Direct and indirect effects of cannabinoids on the nervous and digestive systems are demonstrated (Figure 5).

## 6. Therapeutic Implications and Future Directions

### 6.1. Therapeutic Implications

The burgeoning field of research exploring the therapeutic implications of cannabinoids in neurodegenerative diseases, with a focus on their modulation of central and peripheral immunity, holds substantial promise. Within this context, neuroinflammation, recognized as a pivotal factor in diseases such as AD, PD, and MS, plays a key role in disease progression. Cannabinoids, including CBD and THC, demonstrate anti-inflammatory properties, providing a novel avenue for managing these conditions [245,246]. By interacting with cannabinoid receptors in the endocannabinoid system, these compounds influence immune cell activity, cytokine release, and overall inflammatory responses in central nervous and peripheral tissues. This modulation presents significant potential for mitigating the destructive effects of chronic neuroinflammation and slowing the degenerative processes associated with these diseases [247].

The therapeutic implications of cannabinoids in AD have gained attention due to their potential neuroprotective and anti-inflammatory properties. Cannabinoids, including CBD and THC, have shown promise in preclinical studies by modulating the immune response and reducing neuroinflammation, which are key factors in the disease’s progression. Future directions in research may focus on elucidating the specific mechanisms through which cannabinoids interact with the intricate pathology of AD [248]. Additionally, pre-clinical and clinical trials exploring the efficacy and safety of cannabinoid-based interventions for managing AD symptoms and altering disease progression are crucial steps in advancing potential therapeutic applications [174,249]. As we navigate the complexities of AD, cannabinoids may emerge as valuable tools in the pursuit of innovative treatments that could alleviate symptoms and slow the degenerative processes associated with this debilitating condition [250].

In the realm of PD, cannabinoids have garnered attention due to their potential neuroprotective and anti-inflammatory attributes [245]. Research studies propose that cannabinoids, with a particular emphasis on CBD, may alleviate motor symptoms and mitigate neuroinflammation associated with PD [251,252]. Recent evidence indicates that cannabinoid receptors may play a crucial role in slowing down the advancement of PD by activating neuroprotective pathways [253]. Looking ahead, further exploration into cannabinoid-based therapies may shed light on their ability to regulate the immune response and decelerate degenerative processes in HD [254]. Preliminary investigations into HD suggest cannabinoids exhibit neuroprotective effects and reduce inflammation in the central nervous system, indicating therapeutic potential for addressing motor and cognitive symptoms [255]. Ongoing research endeavors aim to elucidate the mechanisms through which cannabinoids interact with the immune system in HD, opening avenues for targeted therapeutic interventions.

### 6.2. Future Directions

In the case of ALS, cannabinoids have emerged as a key focal point in therapy exploration owing to their anti-inflammatory and neuroprotective properties. Evidence indicates that cannabinoids could modulate the immune response, potentially impeding the degeneration of motor neurons [255,256]. Future directions may encompass clinical trials assessing the effectiveness and safety of cannabinoids in managing ALS symptoms and influencing disease progression. While Multiple System Atrophy (MSA) has received less comprehensive investigation regarding cannabinoids, there exists potential to explore their ability to modulate immune responses [245,257,258]. Subsequent research may delve into whether cannabinoids can alleviate neuroinflammation and enhance autonomic function in individuals with MSA, introducing innovative avenues for therapeutic intervention. In the context of frontotemporal dementia (FTD), the limited research on cannabinoids underscores their anti-inflammatory effects, suggesting potential therapeutic applications [259,260,261]. Future explorations may involve investigating whether cannabinoids can alleviate cognitive and behavioral symptoms by modulating immune responses in affected brain regions, offering fresh insights into potential treatments for FTD.

As we contemplate future directions, a critical aspect is the need for a deeper understanding of the mechanisms underlying cannabinoids’ immunomodulatory effects. Researchers must elucidate intricate signaling pathways and cellular interactions to design targeted and efficacious interventions. Exploring varying impacts across different stages of neurodegenerative diseases and understanding disease-specific effects is crucial for tailoring treatment strategies. Standardized protocols for cannabinoid administration and dosage are imperative for consistency and reliability in therapeutic outcomes, paving the way for evidence-based treatments.

Figure 6 demonstrates possible directions for future studies.

### 6.3. Legal and Ethical Challenges Associated with the Use of Cannabinoids in Clinical Trials

Clinical trials play a pivotal role in determining the practicality and safety of utilizing cannabinoids for therapeutic purposes in neurodegenerative diseases. Rigorous trials are essential to assess short-term efficacy, long-term consequences, and potential side effects of cannabinoid-based interventions. Including diverse patient populations contributes to a comprehensive understanding of therapeutic potential and limitations. Collaboration between researchers, clinicians, and regulatory bodies is crucial to navigating the complex landscape of cannabinoid therapeutics and establishing guidelines for integration into standard treatment protocols. However, performing clinical studies is complicated because of the absence of legal authorization to use cannabinoids for neurodegenerative diseases in many countries.

It is well known that different countries are characterized by various legal bases linked to the use of cannabinoids in clinical practice. The lack of studies where the use of cannabinoids in clinical trials and practice is compared in different countries is problematic. The available facts support an opinion that cannabinoids must be avoided in clinical trials involving children, pregnant women, and people with mental health disorders.

## 7. Conclusions

The increasing acceptance of cannabinoids caused novel preclinical research of neurodegenerative diseases, which was collected and analyzed in this review. These studies demonstrated the neuroprotective properties of many cannabinoids through various cellular and molecular pathways in neurodegenerative diseases. The strengthening connection between the periphery and the CNS in the context of neurodegenerative diseases, together with the extensive immune activities of cannabinoids in both arenas, shows the complexity of immune modulation and the enormous therapeutic potential of cannabinoids in neurodegenerative diseases, which are very difficult to manage.

Multiple additional long-term controlled clinical studies are required for progress in this field.

## Figures and Tables

**Figure 1 ijms-25-06402-f001:**
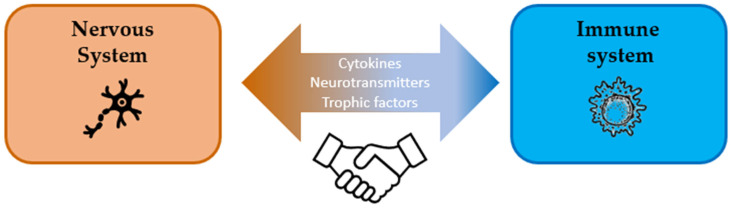
The crosstalk between nervous and immune systems.

**Figure 2 ijms-25-06402-f002:**
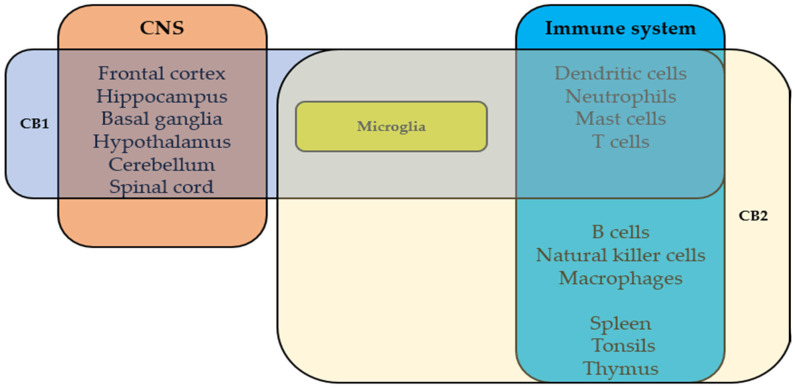
The expression of CB1R and CB2R in nervous and immune systems.

**Figure 3 ijms-25-06402-f003:**
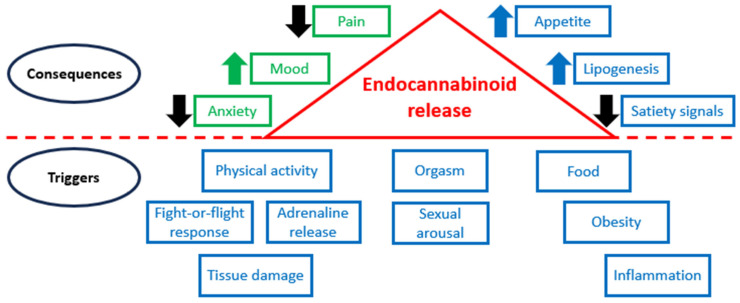
ECS in normal conditions and pathology.

**Figure 4 ijms-25-06402-f004:**
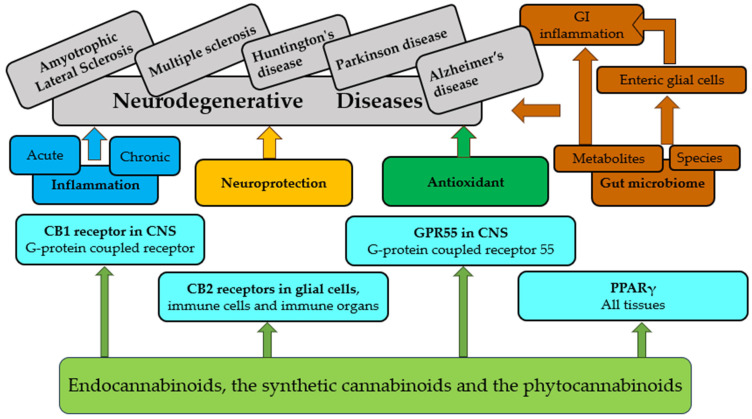
Brief overview of pathways of cannabinoid effects.

**Figure 5 ijms-25-06402-f005:**
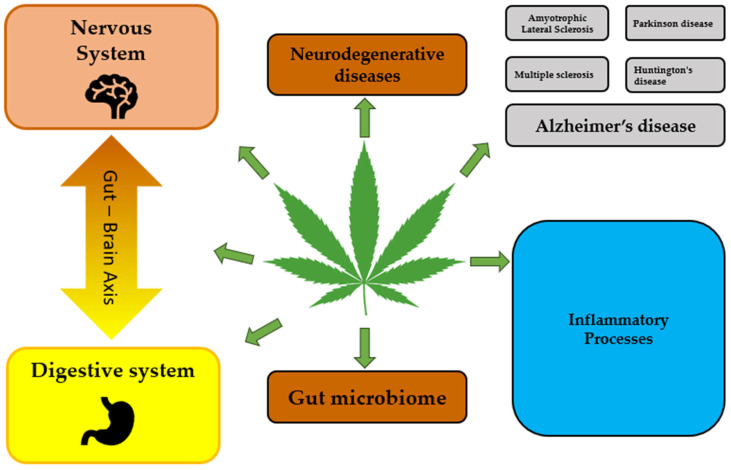
Direct and indirect effects of cannabinoids on nervous and digestive systems.

**Figure 6 ijms-25-06402-f006:**
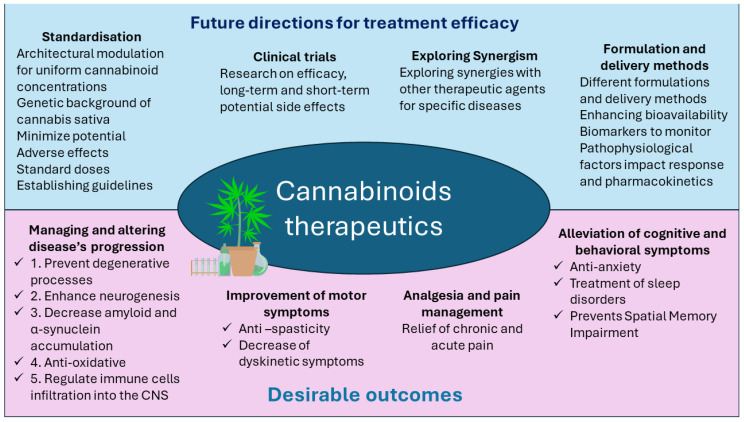
Possibilities of future studies.

**Table 1 ijms-25-06402-t001:** The pharmacological effects of cannabinoids on PD.

Cannabinoids	Model	Effect	Possible Mechanism	References
THC CBD THCV	Animal models	Anti-parkinsonian effects	Antioxidant properties and, in the case of THCV, CB2 activation and CB1 antagonism	[155,156,157]
CBD	Animals: parkinsonian rats	Reduces the orofacial pain threshold	The mechanism is not clear; females and males reacted differently; a hormonal influence is possible	[158]
CP55,940	*Drosophila melanogaster*	Improves locomotor activity	Inactivation of the JNK signaling pathway	[159]
CBD	Clinical study	Improved quality of life	The mechanism is not clear	[153]
THC and nabilone	Clinical study	Reduced levodopa-induced dyskinesia	The mechanism is not clear	[160]
CBD	Clinical study	Decrease in psychotic symptoms	The mechanism is not clear	[154]
Acid cannabinoids	Clinical study	Subjective improvement of motor symptoms	The mechanism is not clear	[161]
Nabilone	Clinical study	Amelioration of anxiety and sleeping problems	The mechanism is not clear	[162]
RCT, nabilone (single oral dose, 0.03 mg/kg)	Clinical study	No effect	The mechanism is not clear	[163]
Oleoylethanolamide (OEA)	6-OHDA model of PD in mice	Decrease in dyskinetic symptoms	Striatal overexpression of FosB and phosphoacetylation of his-tone 3	[164]
WIN-55,212-2	L-DOPA-induced motor fluctuation model of PD	WIN-55,212-2 reduced AIMs to L-DOPA in 6-OHDA-lesioned rats	Modulating DARPP-32 and ERK1/2 phosphorylation in striatal neurons	[165]
THC	In vitro	Neuroprotection	Effect through PPARγ activation	[166]
CBD	In vitro	Neuroprotective effects	Activation of tropomyosin receptor kinase A (TrkA) receptors	[167]
VCE-003.2	In vitro and LPS mouse model	Decreasing the inflammatory response	Mechanism: targets PPARγ	[121]
VCE-003.2	LPS mouse model	Neuroprotective effect	Activation of PPARγ and other signaling pathways	[168]

**Table 2 ijms-25-06402-t002:** The effects of cannabinoids in CNS and periphery in in vivo studies of neurodegenerative diseases.

Cannabinoids	Disease, Model	Effects in CNS	Effects in Periphery	References
10 mg/kg each of THC+CBD (1:1 ratio)	MS, EAE animal models	Reduced disease severity; reduced LPS levels in the brain	Reduced IL-17A and IFN-γ and increased MDSCs in splenocytes; increased short-chain fatty acids in gut microbiome; reduced *Akkermansia muciniphila* in feces	[221]
CBD 20 mg/kg	MS, EAE animal models	Attenuated disease severity, increased MDSCs, and reduced CXCL9, CXCL10, and IL-1β expression	Increased MDSCs and monocytes in the spleen, decreased neutrophils in mesenteric lymph nodes, and suppressed systemic inflammation in the GI tract	[239]
CBD 4.3 mg/kg	PD, transgenic mouse model	Improved motor deficits and prevented αSyn aggregation	Downregulated pathologic metabolites that participate in arginine biosynthesis and histidine metabolism	[233]
CBD 75 mg/kg	MS, EAE animal models	Reduced clinical disease, neuroinflammation in the cerebellum, and T cell infiltration into spinal cord	Suppressed IFN-γ-producing CD8+ T cells in the spleen	[29]
CBD 20 mg/kg	MS, EAE animal models	Attenuated EAE disease progression; MDSCs increased in spinal cord and brain	Reduced IFNγ and IL-17 and increased IL-10 production in the spleen; peritoneal MDSCs elevated	[186]
CB2 agonist, HU-308, 15 mg/kg	MS, EAE animal models	Improved EAE symptoms and reduced spinal cord lesions, microglial activation, and chemokine receptors	Downregulation of chemokines CCL2, CCL3, CCL5 and their receptors CCR2 and CCR1 in bone marrow	[76]
CB2 agonist, Gp1a, 5 mg/kg	MS, EAE animal models	Attenuated EAE development, limited infiltration of CD4 T cells, downregulated pro-inflammatory genes in spinal cords	Suppressed expression of chemokine receptors in splenic T cells	[84]

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
