# Peer review of "Cannabinoids’ Role in Modulating Central and Peripheral Immunity in Neurodegenerative Diseases"

_ijms, 2024, doi:10.3390/ijms25126402_

Round 1

Reviewer 1 Report

Comments and Suggestions for Authors

Cannabinoids, with their potential to influence neuroprotective and anti-inflammatory properties, may play a key role in the future in the treatment of neurodegenerative diseases, with particular emphasis on their modulation of central and peripheral immunity, is extremely promising. In this context, the demonstrated role of CB1R, CB2R and GPR55 receptors is related to inflammation in the nervous system, as a factor in diseases such as Alzheimer's disease (AD), Parkinson's disease (PD) and multiple sclerosis, as well as ALS symptoms and influencing the progression of the disease.

As the authors indicate, future research directions may focus on elucidating the specific mechanisms through which cannabinoids influence the complex pathology of AD, PD and multiple sclerosis. cannabinoids can modulate the immune response, potentially hindering neuronal degeneration. The interactions between the periphery and the CNS in the context of neurodegenerative diseases, together with the extensive immunological activity of cannabinoids in both areas, demonstrates the complexity of immune system modulation and the enormous therapeutic potential of cannabinoids in neurodegenerative diseases that are very difficult to treat. Overall, the manuscript is well written.

There are no major comments regarding the described scope of knowledge about the biological effects of cannabinoids in the neuroprotective and anti-inflammatory context. The literature cited is adequate.

A minor note concerns the length of the manuscript, which may make reading difficult.

Author Response

20 May 2024

Response to Reviewer 1 Comments Manuscript ID: ijms-3016198

"Cannabinoids' Role in Modulating Central and Peripheral Immunity in
Neurodegenerative Diseases"

We kindly thank Reviewer 1 for the time spent reviewing our manuscript, the invested effort, and the constructive comments. We have modified our paper according to the reviewer’s comments and responded in detail to the concerns. We also include the updated version of the manuscript with this letter.

Comments and Suggestions for Authors

  1. There are no major comments regarding the described scope of knowledge about the biological effects of cannabinoids in the neuroprotective and anti-inflammatory context. The literature cited is adequate.

A minor note concerns the length of the manuscript, which may make reading difficult.

 Response: The manuscript was improved and some changes were performed.

The new figures were added, which may make reading less difficult than previously.

Some parts were rewritten in brief form:

In Introduction :

"However, performing clinical studies is complicated because of the absence of legal au-thorization to use cannabinoids for neurodegenerative diseases in many countries. Some clinical research has demonstrated that cannabinoids are promising mediators for the treatment of neurodegenerative diseases, depression, cancer, and so on [5–10], but they have many undesirable side effects, for example, heart disorders, anxiety, possible loss of motor coordination and impaired memory [4,11–13]"

Was changed to

" Futhermore, cannabinoids are promising mediators for the treatment of neurodegenerative diseases, depression, cancer, and so on [5–10] according to clinical studies, but they have many undesirable side effects, for example, heart disorders, anxiety, possible loss of motor coordination and impaired memory [4,11–13]" ;

"To our knowledge, not many reviews are devoted to cannabinoids in modulating central and peripheral immunity" was deleted.

it was added at the beginning of section 2 :

"The endocannabinoid system (ECS) is important multifunctional part of the nervous system and some peripheral tissues; its main components are cannabinoids, cannabinoid receptors, and the various proteins [2, 4, 6]. Its function is connected to numerous processes such as pain, food intake, energy metabolism, inflammation, immune, gastrointestinal, hormonal, and cardiovascular activities and so on [2-6]".

Then, the sentence

"Cannabinoid receptors (CBRs) are part of the endocannabinoid system" was deleted.

"Schmole et al. found that CB2Rs' deletion improved cognitive and learning deficits and decreased microglia activation and amyloid levels in amyloid precursor protein transgenic mice [74]."

Was changed to

"CB2Rs' deletion improved cognitive and learning deficits and decreased microglia activation and amyloid levels in amyloid precursor protein transgenic mice [74]."

In section 2.4

" PPARγ is a nuclear receptor that plays roles in various cellular functions and regulates energy homeostasis and glucose and lipid metabolism"

Was changed to

PPARγ is a nuclear receptor that regulates cellular functions, energy homeostasis, glucose and lipid metabolism" 

It was added (p.8-9):

"  For example, seven patients with PD had significant improvements in measures of functioning and well-being after treatment with CBD 300 mg/day compared to a placebo group [159]; it was reported that CBD decreased psychotic symptoms [161].

The part in section 6 was deleted:

"Exploring advanced delivery methods and formulations is another avenue for future exploration. Enhancing bioavailability and precision of cannabinoid administration could optimize therapeutic impact and minimize potential adverse effects. This includes investigating novel drug delivery systems, exploring synergies with other therapeutic agents, and identifying biomarkers to monitor treatment response. Addressing these challenges and refining understanding of cannabinoids' immunomodulatory effects positions the field for significant advancements that could revolutionize the therapeutic landscape for neurodegenerative diseases, making cannabinoids versatile tools in pursuing effective treatments."

Reviewer 2 Report

Comments and Suggestions for Authors

I would like to show my sincere greeting to the authors for their valuable work.

I see that the review needs to be better structured and well-written; more figures and illustrations would improve the presentation; additionally, I suggest including more details on the reported findings of previous clinical research.

Comments on the Quality of English Language

The English language needs revision.

Author Response

20 May 2024

Response to Reviewer 2 Comments Manuscript ID: ijms-3016198

"Cannabinoids' Role in Modulating Central and Peripheral Immunity in
Neurodegenerative Diseases"

We kindly thank Reviewer 2 for the time spent reviewing our manuscript, the invested effort, and the constructive comments. We have modified our paper according to the reviewer’s comments and responded in detail to the concerns. We also include the updated version of the manuscript with this letter.

Comments and Suggestions for Authors

I see that the review needs to be better structured and well-written; more figures and illustrations would improve the presentation; additionally, I suggest including more details on the reported findings of previous clinical research.

Response: it was performed, the manuscript was edited, new figures were added, and more details of clinical studies were added.

Reviewer 3 Report

Comments and Suggestions for Authors

This is a review article by Sharon et al on the potential Role of Cannabinoids' in Modulating Central and Peripheral Immunity in Neurodegenerative Diseases. Overall, authors have done a good job in reviewing the existing literature but I have a few comments that could potentially benefit this manuscript:

1. Suggest authors to include a search strategy (what terms were used, which search engines were used, any inclusion/exclusion criteria etc.), if authors did not employ any specific search strategies, the same should still be included in the Introduction section.

2. Suggest authors to first elaborate on the endocannabinoid system (ECS) and its role  in general before discussing its receptors.

3. Strongly suggest authors to include/create a separate sub-section on the legal and ethical challenges associated with the use of cannabinoids in clinical practice and clinical trials; including variance by country.

4. May be I missed it, but I did not see Table 2 cited in the text.

Comments on the Quality of English Language

Quality of English language is acceptable.

Author Response

20 May 2024

Response to Reviewer 3 Comments Manuscript ID: ijms-3016198

"Cannabinoids' Role in Modulating Central and Peripheral Immunity in
Neurodegenerative Diseases"

We kindly thank Reviewer 3 for the time spent reviewing our manuscript, the invested effort, and the constructive comments. We have modified our paper according to the reviewer’s comments and responded in detail to the concerns. We also include the updated version of the manuscript with this letter.

Comments and Suggestions for Authors

  1. Suggest authors to include a search strategy (what terms were used, which search engines were used, any inclusion/exclusion criteria etc.), if authors did not employ any specific search strategies, the same should still be included in the Introduction section.

Response: it was added in the Introduction section:

"Several databases such as PubMed, Google Scholar, Scopus and Science Direct were examined using the key words; post-2001 publications were included which reflect the main facts, ideas, and concepts of the review".  

  1. Suggest authors to first elaborate on the endocannabinoid system (ECS) and its role  in general before discussing its receptors.

Response: it was added in the beginning of the section 2 :

"The endocannabinoid system (ECS) is important multifunctional part of the nervous system and some peripheral tissues; its main components are cannabinoids, cannabinoid receptors, and the various proteins [2, 4, 6]. Its function is connected to numerous processes such as pain, food intake, energy metabolism, inflammation, immune, gastrointestinal, hormonal, and cardiovascular activities and so on [2-6]. Figure 2 illustrates that physical activity, stresses, food consumption, sexual behavior, orgasm, obesity, inflammation, tissue damage and other stimuli  may trigger the release of endocannabinoids, their high levels cause various consequences."

Figure 2 was added.

  1. Strongly suggest authors to include/create a separate sub-section on the legal and ethical challenges associated with the use of cannabinoids in clinical practice and clinical trials; including variance by country.

Response: it was added subsection:

6.1. Legal and ethical challenges associated with the use of cannabinoids in clinical trials

Clinical trials play a pivotal role in determining the practicality and safety of utilizing cannabinoids for therapeutic purposes in neurodegenerative diseases. Rigorous trials are essential to assess short-term efficacy, long-term consequences, and potential side effects of cannabinoid-based interventions. Including diverse patient populations contributes to a comprehensive understanding of therapeutic potential and limitations. Collaboration between researchers, clinicians, and regulatory bodies is crucial to navigating the complex landscape of cannabinoid therapeutics and establishing guidelines for integration into standard treatment protocols. However, performing clinical studies is complicated because of the absence of legal authorization to use cannabinoids for neurodegenerative diseases in many countries.

It is well known that different countries are characterized by various legal base linked to the use of cannabinoids in clinical practice. The lack of studies where the use of cannabinoids in clinical trials and practice is compared in different countries is problematic. The available facts support an opinion that cannabinoids must be avoided in clinical trials involving children, pregnant women, and people with mental health disorders.

  1. May be I missed it, but I did not see Table 2 cited in the text.

Response: it was added at the end of the section 5 :

"Table 2 summarizes the experimental data of neurodegenerative diseases in animal models".

Round 2

Reviewer 2 Report

Comments and Suggestions for Authors

I would like to thank the authors for their effort and fast response, good changes were made but I still see that the topic needs to be illustrated with more descriptive and significant figures, and the information provided in the manuscript needs to be more organized.

Author Response

30 May 2024

Response to Reviewer 2 Comments Manuscript ID: ijms-3016198

"Cannabinoids' Role in Modulating Central and Peripheral Immunity in

Neurodegenerative Diseases"

We kindly thank Reviewer 2 for the time spent reviewing our manuscript, the invested effort, and the constructive comments. We have modified our paper according to the reviewer’s comments and responded in detail to the concerns. We also include the updated version of the manuscript with this letter.

Comments and Suggestions for Authors

I would like to thank the authors for their effort and fast response, good changes were made but I still see that the topic needs to be illustrated with more descriptive and significant figures, and the information provided in the manuscript needs to be more organized.

Response: it was performed. Two more figures were added (a total of six illustrations and two tables), and we have reorganized the review by dividing it into leading subchapters.
